# Nurse resilience, burnout, pandemic stress, and post-traumatic stress: A secondary analysis of a longitudinal cohort

Rachel G. Baskin[1,2]*, Linda C. Copel[1], Janell L. Mensinger[3], Heather Brom[4]

**1** M. Louise Fitzpatrick College of Nursing, Villanova University, Villanova, Pennsylvania, United States of America, **2** School of Nursing, Widener University, Chester, Pennsylvania, United States of America, **3** College of Psychology, Nova Southeastern University, Fort Lauderdale, Florida, United States of America, **4** School of Nursing and Leonard Davis Institute of Health Economics, University of Pennsylvania, Philadelphia, Pennsylvania, United States of America

\* rgbaskin1@widener.edu

## Abstract

### Background

It is estimated that approximately one-fifth of nurses in the United States will leave the profession by 2027 due to stress and burnout caused by the COVID-19 pandemic. It is unknown how burnout, resilience, and post-traumatic stress changed during the first two years of the COVID-19 pandemic in frontline nurses. The primary aim of this study was to evaluate how resilience, burnout, and post-traumatic stress changed in hospital-based nurses from 2020 to 2022. Secondary objectives were to describe the relationships between them and test whether burnout and resilience mediated the relationship between pandemic stress and post-traumatic stress.

### Methods

This was a secondary analysis of a longitudinal cohort study of hospital nurses who participated in the COVID-19 Study and Registry of Healthcare and Support Personnel (CHAMPS) Registry. Changes in resilience, burnout, and post-traumatic stress (PTS) were evaluated using repeated measures ANOVA. Path analysis was conducted using multiple regressions to identify whether burnout and resilience acted as mediators between pandemic stress and post-traumatic stress.

### Results

Thirty-two participants were included in all four waves of the longitudinal study, with a range of 32 to 740 participants across all time points. Changes in PTS were significant, while changes in burnout and resilience were not. Eighty-nine participants were available for the regression models used to answer the secondary objectives. Burnout mediated the relationship between pandemic stress and post-traumatic stress,

**Data availability statement:** Data Availability: Data cannot be shared publicly because of the risk of loss of confidentiality to participants. Data are available from the Villanova University Institutional Review Board for researchers who meet the criteria for access to confidential data. Deidentified CHAMPS data and codebooks

are available upon request from the Villanova University Fitzpatrick College of Nursing. Please contact the study PI, Dr. Margaret Brace (margaret.brace@villanova.edu).

**Funding:** This work received funding from Villanova University's Falvey Memorial Library Scholarship Open Access Reserve (SOAR) Fund and Villanova University's Alpha Nu Chapter of Sigma Theta Tau International. There was no additional external funding received for this secondary study. The primary study was supported by grants from Travere Therapeutics Inc, McKesson Corporation, anonymous donors, and internal funding from the M. Louise Fitzpatrick College of Nursing at Villanova University.

**Competing interests:** The authors have declared that no competing interests exist.

but resilience did not. In addition, adequate protective equipment was found to be a predictor of lower pandemic stress.

## Conclusions

Post-traumatic stress peaked in 2020 during lockdown in the United States and decreased significantly by 2022. Resilience and burnout did not change between 2020 and 2022. The results of this study can guide healthcare organizations in providing frontline healthcare workers with mental health resources, especially at the outset of a pandemic.

---

## Introduction

Even beyond the pandemic's peak, COVID-19 continued to cause psychological distress among frontline healthcare workers [1,2]. The burden of the pandemic has negatively affected healthcare workers' mental health and overall well-being, including increased burnout, decreased resilience, and increased post-traumatic stress [3,4]. The National Academy of Medicine's (NAM's) *Future of Nursing 2020−2030* report anticipated over the next decade that there will be long-term effects from the COVID-19 pandemic on the nursing workforce, although specific long-term effects were not described [4]. Additionally, NAM highlights in their *National Plan for Health Workforce Well-Being* the need for research regarding the immediate and long-term effects of the COVID-19 pandemic on healthcare workers' well-being [5].

Previous studies have linked burnout and other indicators of nurses' mental health to a variety of patient outcomes including hospital acquired infections, falls, pressure ulcers, and medication errors [6,7]. Given that nurses (RNs) represent the largest professional healthcare workforce, it is important to identify the long-term effects that working during the pandemic has had on their mental health and overall well-being [6]. In a profession already facing staffing shortages, healthcare systems must prioritize nurses' well-being to develop effective interventions and retain nurses [8,9].

While over 40 cross-sectional studies measured resilience, burnout, and/or post-traumatic stress (PTS) at various time points during the COVID-19 pandemic, only eight examined these outcomes longitudinally [10–17]. Among these eight studies, a mere three focused on nurses, one of which was a repeated cross-sectional study of nurses at three time points [11–13]. In one of these studies, frontline nurses in China experienced significantly worse post-traumatic stress during the "outbreak period" from January to February 2020 (33%) compared to the "stable period" two-weeks later (19.1%) [11]. In a repeated cross-sectional study of 228–320 nurses in the United States over three time points from April 2020 to April 2021, the prevalence of severe burnout was 30% and 14% at the six-month and one-year follow up periods, respectively [12]. Satisfaction with personal protective equipment (PPE) availability was inversely and weakly correlated with burnout [12]. Finally, in a study of 169 Taiwanese emergency nurses, there was no significant difference in burnout and suspected post-traumatic stress disorder (PTSD) at baseline (July-August 2020)

and three-month follow-up (October-November 2020) [13]. Approximately 41% of nurses were experiencing six or more symptoms of post-traumatic stress at baseline, compared to 33% at the three-month follow-up [13]. Taken together, these studies demonstrated that well-being outcomes, including burnout and post-traumatic stress, were generally worse during baseline assessments compared to the follow-up assessments [11–13].

What remains unknown is how resilience and burnout changed beyond the first year of the pandemic and whether these constructs mediated the severity of PTS experienced by frontline nurses. Mediation shows how one variable (e.g., burnout) explains the relationship between a predictor and an outcome [18]. These analyses are crucial for understanding individual variations in pandemic-related mental health outcomes [18]. Understanding the longitudinal effects of working during a pandemic on the mental health of hospital-based nurses and potential mediators can guide healthcare organizations in implementing proactive measures for frontline workers during future pandemics. Using longitudinal data from the COVID-19 Study of Healthcare and Support Personnel (CHAMPS) Registry, this secondary analysis examined measures of well-being in nurses across four time points from 2020 to 2022 [19].

### Conceptual model

An adapted version of the Chronic Traumatic Stress (CTS) model guided this study. The CTS was originally created to highlight the diverse responses to chronic stress among refugees and victims of war crimes [20]. Stressful events, including past and current war trauma, daily stressors, and post-migration living difficulties, were predictors in the original model. These stressful events can lead to various outcomes, such as psychological and physical distress, as well as post-traumatic stress symptoms. Protective factors (e.g., age, sex, resilience, spirituality, coping style) and risk factors (e.g., substance abuse, domestic violence, community violence) were suggested moderators of the impact of stressful events on well-being. The predictors, outcomes, and moderators listed in the CTS are not comprehensive and were based on the available literature [20]. The CTS model is currently the only conceptual model that attempts to describe the effects of chronic stress after exposure to a traumatic event.

### Adapted CTS model

For this study, the CTS model was adapted to evaluate the long-term effects that working during the first two years of the COVID-19 pandemic had on frontline nurses. Exposure to stressful events has been associated with post-traumatic stress symptoms in healthcare workers; therefore, PTS was the outcome variable in the model [21,22]. In a study measuring pandemic-related stress during the SARS outbreak, high pandemic-related stress was associated with a high level of PTS symptoms (OR= 5.6, $p < .001$) [22]. Therefore, an adapted measure of pandemic stress was used as a predictor in the adapted CTS model because it could describe the severity of stress experienced by frontline nurses, and the outcome of interest was PTS during one-year follow-up. Although resilience was an individual protective factor (moderator) in the original CTS, it was included as a mediator in the adapted model due to several studies of nurses conducted during the COVID-19 pandemic that described resilience as a mediator between variables of stress and/or burnout [24–26]. Using the adapted CTS model, we posited that resilience and burnout would mediate the severity of post-traumatic stress symptoms in nurses during the COVID-19 pandemic. Based on prior literature, burnout was included as a mediator since it was a precursor to PTS and because burnout also mediated the relationship between stress and post-traumatic stress before and during the COVID-19 pandemic [13,27,28].

Due to the potential confounding effects of antecedents on the variables of interest, we proposed they be added to the adapted model. Antecedent variables occur before the independent and dependent variables and can affect the relationship between the two [29]. Therefore, antecedents to pandemic-related risk were added to the model, including prior trauma and adequacy of work resources. It was hypothesized that prior trauma and inadequate resources would have a negative influence on nurses' pandemic-related stress [30,31]. Adequate work resources in this study included personal protective equipment (PPE), staffing, equipment (e.g., ventilators), and preparations for an influx of patients with COVID-19.

## Study purpose

Informed by the Chronic Traumatic Stress model, the purpose of this secondary analysis of longitudinal data was to describe how resilience, burnout, and post-traumatic stress changed during the pandemic and how these concepts were related. Using data from nurses collected through the CHAMPS Registry across four time points (T1 baseline, T2 six-month follow-up, T3 one-year follow-up, and T4 two-year follow-up), the specific aims of this study were to:

**Aim 1:** Identify how resilience, burnout, and post-traumatic stress changed over time from T1 (2020) to T4 (2022).

**Aim 2:** Test the relationships between pandemic stress at T1 (independent variable), resilience and burnout at T2 (mediators), and post-traumatic stress at T3 (dependent variable) in hospital-based nurses during the COVID-19 pandemic.

## Materials and methods

### Study design

This study was a secondary analysis based on a longitudinal cohort design. It analyzed data from the CHAMPS Registry to determine how burnout, resilience, and post-traumatic stress changed over time in nurses during the COVID-19 pandemic from 2020 to 2022.

### CHAMPS registry

The CHAMPS survey and registry (NCT04370821) was led by researchers at Villanova University and designed to evaluate the longitudinal effects of the pandemic on essential workers' mental and physical well-being [19]. The CHAMPS registry collected data on a variety of essential personnel roles, the majority of whom were nurses, and included other healthcare workers, first responders, and housekeeping staff. The CHAMPS study was conducted in the U.S. using snowball sampling and advertising via local and national professional organizations. Social media was also used to distribute the link to the study consent form and questionnaires [19]. Recruitment took place for 13 months from May 2020 to June 2021 (T1). A total of 2,762 participants enrolled at T1, 1,534 of whom agreed to be contacted as part of the longitudinal study. The CHAMPS study employed questionnaires at four different time points: 1) May 2020- June 2021 (T1); 2) November 2020 to May 2021 (T2); 3) May 2021 to May 2022 (T3); and 4) May to June 2022 (T4). Due to low participation at T4, the CHAMPS researchers closed the survey after two months of data collection. A detailed description of the CHAMPS protocol can be found in Kaufmann et al. [19]

### Setting and sample

For this study, the sample included registered nurses who reported they worked in an inpatient hospital setting and cared for patients with suspected or confirmed COVID-19 at T1. Nurses who identified themselves as working in a supervisory or managerial role were excluded. Of the 1,027 registered nurses who participated at T1, 811 met the inclusion criteria for this study. However, 71 participants did not respond to the questions related to the variables of interest at T1 and were excluded from analysis, leaving a total of 740 participants in the sample. For the longitudinal component of this study, 32 participants responded to all four questionnaires. Approximately 95% of the sample dropped out between T1 and T4. The number of participants and variables measured at each data collection point are listed in **Fig 1**.

### Measures

**Pandemic stress.** Pandemic stress was measured in the baseline survey only (T1) using the Stress Specific to COVID-19 scale. Stress Specific to COVID-19 is conceptualized as a person's perception of stress, particularly while working in a high-risk environment during a pandemic [22,23]. Specific stressors related to the COVID-19 pandemic were adapted from risks assessed during the SARS pandemic, such as fear of getting sick, not surviving the illness, or passing

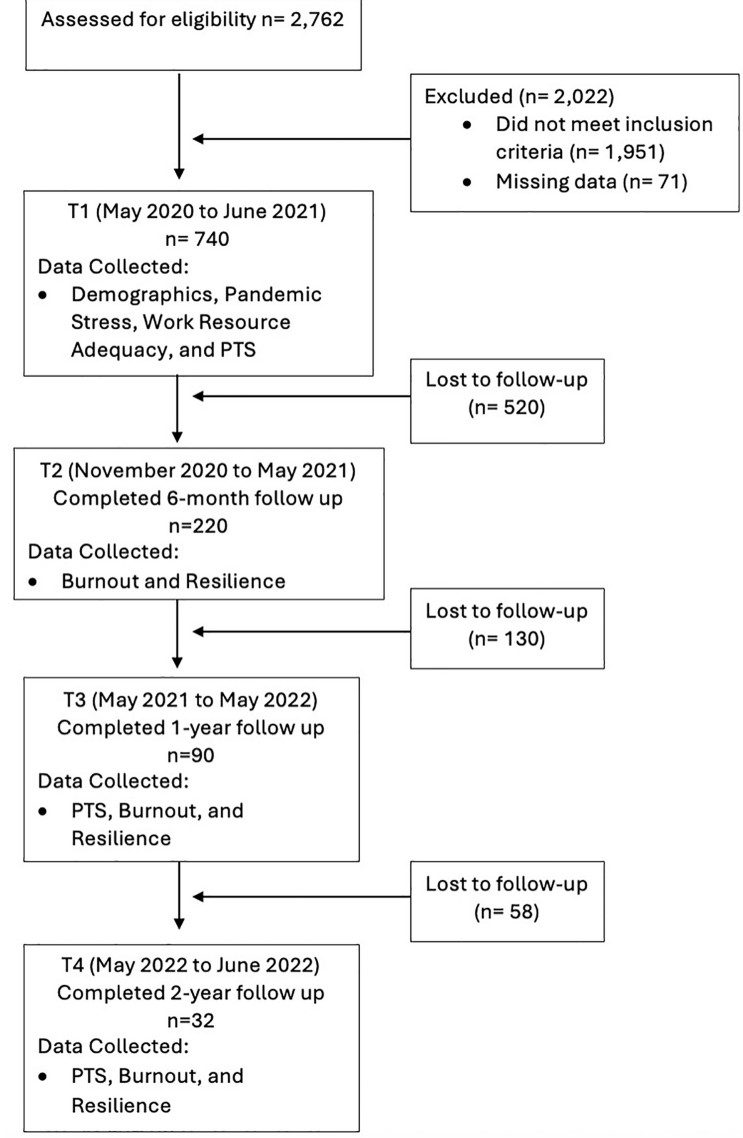

**Fig 1. Variables, the Time Points Collected, and the Number of Nurses Who Responded.**

it on to others [22,23]. For this study, nine items addressing perceptions of COVID-19-related risks were measured on a five-point Likert scale from strongly agree (1) to strongly disagree (5). Example items included, "I feel extra stress at work" and "I believe my job is putting me at great risk" [22]. A composite score was created based on the average of all items. Wu et al. reported a Cronbach's alpha value of .71, and pandemic stress was significantly associated with post-traumatic stress, demonstrating construct validity [22]. The Cronbach's alpha in this study was .77.

**Burnout.** Burnout was measured three times during this study at six-months, one-year, and two-years (T2, T3, and T4, respectively). According to the World Health Organization, burnout is a phenomenon that occurs due to chronic stress in the workplace [32]. Burnout was measured using the Oldenburg Burnout Inventory (OLBI) [33]. The OLBI is a 16-item instrument made up of two scales: exhaustion and disengagement from work [33]. Exhaustion is defined as the

consequence of intense physical and cognitive strain, and disengagement is defined as "distancing oneself from work and experiencing negative attitudes towards work" [34] (p.14)]. Participants are asked the extent to which they agree or disagree with statements about feeling emotionally drained or disconnected at work using a four-point Likert scale. Total scores range from 16 to 64. A score of <44 is categorized as low burnout, scores of 44−59 are categorized as moderate burnout, and scores >59 as high burnout [35]. The internal consistencies for the subscales and total scores were acceptable at all time points, with Cronbach alpha scores of .74 to .90 [33]. The English version of the OLBI was found to be reliable and valid using confirmatory factor analysis and comparison to the Maslach Burnout Inventory- General Survey [33]. The Cronbach's alpha for the total burnout scores in this study were acceptable (.86− .91).

**Resilience.** Resilience was measured concurrently with burnout at six-months, one-year, and two-years (T2, T3, and T4, respectively). The Brief Resilience Scale (BRS) was used to measure an individual's resilience, or ability to bounce back [36]. The BRS is a six-item scale that asks participants to rate statements about recovering from stressful events using a five-point Likert scale from strongly disagree (1) to strongly agree (5), with total scores calculated by the mean of all responses [36]. A mean of 3.00 and below is interpreted as low resilience and a mean of 4.30 and above as high resilience [37]. The BRS had good internal consistency (Cronbach's alpha range of .80− .91) and test-retest reliability with intraclass correlations of .69 for one month and .62 for three months [36]. The Cronbach's alpha values in this study were .86 − .87 across the three data collection points.

**Post-traumatic stress.** Post-traumatic stress was measured in the baseline survey (T1) and at the one-year and two-year follow-ups (T3 and T4, respectively). The Impact of Events Scale (IES-R) is a 22-item self-report tool that asks about distress caused by the symptoms of post-traumatic stress, rather than their frequency [38]. By measuring the severity of symptoms through self-reported ratings of distress, differentiation can occur between patterns of normal recovery and disordered recovery [38]. The IES-R measures three constructs: intrusion, avoidance, and hyperarousal [39]. Intrusion includes intrusive thoughts, feelings, or nightmares; avoidance is the numbing of responsiveness and avoidance of feelings, situations, and/or ideas related to the traumatic event; and hyperarousal is related to feelings of anger, irritability, hypervigilance, and heightened startle [39]. Participants are asked to rank how distressed they have been in the last seven days related to difficulties occurring after exposure to a traumatic event, such as having trouble sleeping or concentrating, not talking or thinking about it, or having dreams or feelings about it. The questions are asked using a five-point Likert scale from 0 (not at all) to 4 (extremely), with a total score of 0–88. Higher scores cause greater concern for the presence of PTSD, with a suggested cutoff score of 25 for partial PTSD and 33 for probable diagnosis of PTSD; therefore a cutoff of 25 was used in this study [40,41]. The IES-R had high internal consistency for the total score (Cronbach's alpha = .96) and the three subscales (intrusion = .94, avoidance = .87, hyperarousal = .91) [41]. The Cronbach's alpha values in this study for the total scores were .93 to .95 across the three data collection points.

**Antecedents.** Antecedents to pandemic stress were measured at T1 and consisted of prior exposure to a traumatic event, adequate staffing, adequate PPE, and adequate equipment. Prior exposure to a traumatic event was retrospective in nature, so it naturally was an antecedent to pandemic stress at T1. In addition, adequate work resources were selected as antecedents to pandemic stress due to previous studies linking work resources (e.g., infection control processes, adequate staffing, PPE availability) to mental health outcomes (e.g., stress, burnout, PTS) in healthcare workers [12,42].

Prior exposure to a previous traumatic event was asked as a "yes" or "no" question. To measure the remaining four antecedents, participants were asked the extent to which they agreed or disagreed with the following statements: 1) "staffing levels are adequate for the current number of patients being treated in my workplace," 2) "we have adequate personal protective equipment to reduce risk of infection in my workplace," 3) "we have adequate ventilating devices and other life-saving equipment to handle the current number of patients in my workplace," and 4) "my workplace has made adequate preparations for a potential influx of COVID-19 patients in the future." Each item was measured on a seven-point Likert scale from strongly disagree (1) to strongly agree (7) and average scores of each item were used in the analysis. The proposed path model can be found in **Fig 2**.

## Ethical considerations

The primary study was approved by the Institutional Review Board (IRB) at Villanova University in March 2020 (IRB-FY2020–215). This secondary study was approved as exempt by the IRB at Villanova University in March 2024 and therefore, informed consent was waived (IRB-FY2024–159). Deidentified data were accessed in April 2024 and analyzed between April and September 2024.

## Data analysis

Data were analyzed using the Statistical Package for the Social Sciences (SPSS) version 29 and SPSS Process Macro (version 4.3.1). Descriptive statistics were used to describe the sample's demographic characteristics, including age, sex, U.S. region, practice unit type, education level, and years of experience. Mean scores of burnout, resilience, and PTS were grouped into low, medium, and high scores based on suggested cutoffs in the literature [35,37,40,41]. Since the enrollment survey at T1 was open for 13-months, we analyzed the differences in mental health outcomes and work resources between those who responded in the first six-months of T1 to those who responded in the second six-months of T1.

To analyze how burnout, resilience, and post-traumatic stress changed over three time points during the pandemic (Aim 1), a one-way, repeated measures analysis of variance (ANOVA) was conducted using the sample of 32 participants who responded to all four surveys. Post hoc testing using pairwise comparisons was performed to identify significant differences between survey periods, and a Bonferroni adjustment was performed. The sphericity assumption was met for all three variables (S1 Table). For Research Aim 2, SPSS Process Macro Model 4 was utilized to evaluate whether burnout or resilience mediated the relationship between pandemic stress and post-traumatic stress experienced by frontline nurses. Using a 95% confidence interval and bootstrapping with 5,000 samples, the mediation effects were tested. The first regression model used the antecedents of prior exposure to a traumatic event and adequate work resources at T1 as the predictor variables and pandemic stress at T1 as the outcome variable to evaluate if work resources or prior trauma influenced pandemic stress. In

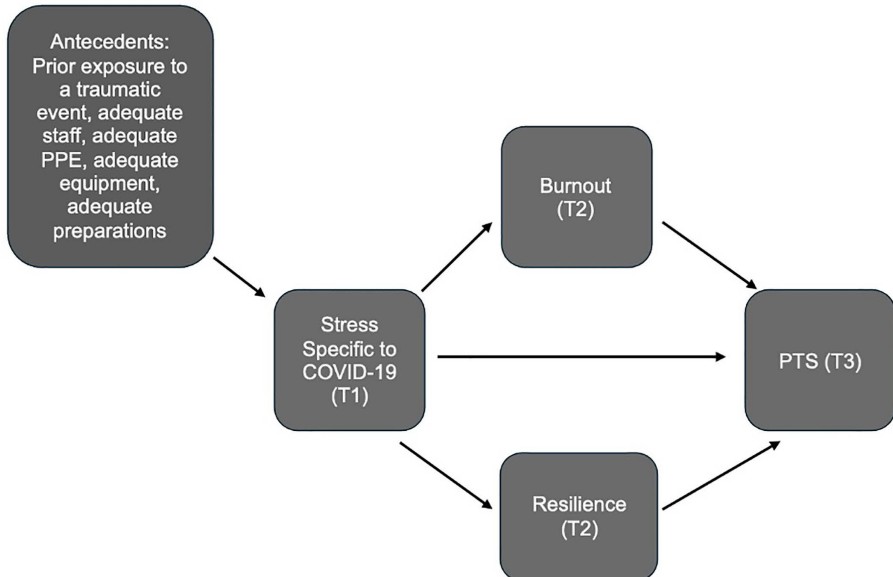

**Fig 2. Proposed path model of the effects of prior trauma, adequate work resources, burnout and resilience on stress specific to COVID-19 and PTS.**

the second regression model, pandemic stress at T1 was the predictor variable and burnout and resilience at T2 were the outcome variables (serving as path a of the mediation model). For the third regression model, the direct effect of pandemic stress (predictor) on PTS at T3 (outcome) was tested (path c of the mediation model). In the final regression model, pandemic stress at T1 (path c' of the mediation model), and burnout and resilience at T2 (path b of the mediation model) were the predictor variables and post-traumatic stress at T3 was the outcome variable.

### Power analysis

Using G*Power, an a priori power analysis was conducted using an alpha of .05, power of 0.8, and medium effect size (f = .25), the required sample size to power Research Aim 1 was n = 28. For Research Aim 2 using an alpha of .05, power of 0.8, and medium effect size (f2 = .15), a minimum sample of 85 was required. Therefore, the sample sizes of 32 for Aim 1 and 89 for Aim 2 were adequate.

## Results

### Sample

The study sample consisted of 740 hospital nurses caring for COVID-patients at T1. They were primarily female (n = 676, 91.4%) and white (n = 651, 88.0%) and were on average 37.0 years old (SD = 12.0) with an average of 11.1 years' experience (SD = 11.0). Participant demographic characteristics are displayed in Table 1. Since the enrollment period for T1 was 13-months long and pandemic conditions experienced a lot of fluctuation, responses on key study variables were compared between those who completed T1 in the first half of the enrollment period (May 2020 to November 2020) and those who completed T1 in the second half of the enrollment period (December 2020 to May 2021; S2 Table). We found no significant differences in pandemic stress, burnout (at T2), resilience (at T2), and PTS between participants who completed T1 during the first half and second half of the enrollment period. There were significant differences in all items about adequate work resources, with better staffing, equipment, and preparations more likely to be reported during the first six months of the pandemic (p values between < .001 and .025). Adequate PPE was more likely to be reported during the second half of T1 (December 2020 to May 2021; p < .001). For most demographics, there were no statistically significant differences in those who completed the first two questionnaires compared to those who completed all four questionnaires. Participants who remained in the CHAMPS study were more frequently employed in the intensive-care unit (ICU) (p = .005) and more frequently reported more adequate staffing at work (p = .037) (S3 Table).

Approximately half (57.6%) of participants (n = 740) agreed that their workplace had adequate staffing compared to the current patient volume at T1. More than half of participants reported having adequate PPE to reduce infection risk (62.3%), and 79.8% of participants reported having adequate lifesaving equipment. Lastly, 73.0% of participants reported that their workplace was adequately prepared for potential increase in volume of COVID-19 patients. Responses to the adequate work resources questions from T1, which served as antecedents in the path model, are reported in Table 2.

### Changes in burnout, resilience, and PTS (Aim 1)

**Burnout.** During T2 (November 2020 to May 2021), nurses had a mean score of 42.4 on the OLBI, which was just below the threshold of 44.0 for moderate burnout. At T3 and T4, the mean burnout scores were 42.5 and 40.8, respectively. There were 41.0% of nurses that met the cutoff for burnout at T2, which decreased to 38.9% at T3, then increased slightly to 40.6% at T4. The number of participants that were experiencing low, moderate, or high burnout and resilience, or met the cutoff scores of post-traumatic stress at each time-point can be found in Table 3. The results of the repeated measures ANOVA showed that burnout did not significantly change over time, with F (2, 62) = 2.63, p = .080, though time explained 7.8% of the variance in changes in burnout, representing a medium effect size (Table 4).

**Table 1. Participant demographics, Measured at T1 (May 2020- June 2021).**

| Variable | Category | N (%) |
|---|---|---|
| Gender | Female | 676 (91.4) |
| | Male | 64 (8.6) |
| Age (years) | 18-29 | 270 (36.5) |
| | 30-44 | 273 (36.9) |
| | 45-59 | 150 (20.3) |
| | 60 and over | 47 (6.4) |
| Race | Black/African American | 21 (2.8) |
| | Latinx/Hispanic | 24 (3.2) |
| | White/Non- Hispanic | 651 (88.0) |
| | Asian/Pacific Islander | 23 (3.1) |
| | Multi-Racial/ Other | 21 (2.9) |
| Marital Status | Single | 282 (38.3) |
| | Married/Domestic Partnership | 401 (54.4) |
| | Divorced/Separated/Widowed | 54 (7.3) |
| Highest Level of Education | Associates, Diploma, or Technical Degree | 76 (10.3) |
| | Bachelor's Degree | 561 (75.8) |
| | Master's or Doctoral Degree | 103 (13.9) |
| Geographic Region* | Northeast | 414 (56.0) |
| | Midwest | 168 (22.7) |
| | West | 67 (9.1) |
| | South | 90 (12.2) |
| Healthcare Facility Type | Metropolitan Hospital | 395 (53.4) |
| | Suburban/Regional Hospital | 255 (34.5) |
| | Rural/Community Hospital | 90 (12.2) |
| Years of Work Experience | 0-5 | 324 (44.0) |
| | 6-10 | 159 (21.6) |
| | 11-20 | 112 (15.2) |
| | 21-30 | 74 (10.0) |
| | 31-45 | 68 (9.2) |
| Unit Type | Inpatient COVID-19 Unit | 183 (24.7) |
| | Emergency Department/Intensive Care | 309 (41.8) |
| | Medical-Surgical | 98 (13.2) |
| | Procedural** | 51 (6.8) |
| | OB/Maternity | 46 (6.2) |
| | Other*** | 53 (7.1) |
| Employment Status | Full-time (>35 hours weekly) | 612 (82.8) |
| | Part-time | 127 (17.2) |
| Previous Traumatic Event | Yes | 373 (50.4) |
| | No | 367 (49.6) |

*Geographic regions based on the U.S. Census Bureau **Procedural areas include Operating Room, Post-Anesthesia Care Unit, Cardiac Catheterization Lab, Interventional Radiology, etc. ***Other units include COVID-19 Screening Site, Outpatient/Urgent Care, Psychiatric/Behavioral Health, "Other" (not otherwise specified), etc.

**Table 2. Descriptive Statistics of Stress Specific to COVID-19, Burnout, Resilience, PTS, and Adequate Work Resources.**

| Measure | T1 Mean (SD) N=740 | T2 Mean (SD) N=220 | T3 Mean (SD) N=90 | T4 Mean (SD) N=32 |
|---|---|---|---|---|
| Stress Specific to COVID-19 | 3.4 (.7) | – | – | – |
| Burnout, Total | – | 42.4 (5.5) | 42.5 (6.5) | 40.8 (5.7) |
| *Disengagement* | – | 21.0 (3.6) | 21.1 (3.9) | 20.1 (3.4) |
| *Exhaustion* | – | 21.4 (2.5) | 21.4 (3.1) | 20.8 (2.8) |
| Resilience | – | 3.4 (.7) | 3.3 (.7) | 3.3 (.6) |
| Post-Traumatic Stress, Total | 28.8 (17.4) | – | 25.9 (18.8) | 16.8 (13.5) |
| Adequate Work Resources | 4.8 (1.4) | – | – | – |
| *"Staffing levels are adequate for the current number of patients being treated in my workplace."* | 4.3 (2.0) | – | – | – |
| *"We have adequate personal protective equipment to reduce risk of infection in my workplace."* | 4.5 (1.9) | – | – | – |
| *"We have adequate ventilating devices and other lifesaving equipment to handle the current number of patients in my workplace."* | 5.4 (1.4) | – | – | – |
| *"My workplace has made adequate preparations for a potential influx of COVID-19 patients in the future."* | 5.1 (1.7) | – | – | – |

A – indicates data for the measure was not collected for the survey period.

**Table 3. Scores of Burnout, Resilience, and PTS Over Time.**

| Measure | Survey Period | Level | N (%) |
|---|---|---|---|
| Burnout | T2 | Low (< 44) | 130 (59.1) |
| | | Moderate- High (44–88) | 90 (40.9) |
| | T3 | Low (< 44) | 55 (61.1) |
| | | Moderate- High (44–88) | 35 (38.9) |
| | T4 | Low (< 44) | 19 (59.4) |
| | | Moderate-High* (44–59) | 13 (40.6) |
| Resilience | T2 | Low (≤ 3.0) | 67 (30.5) |
| | | Moderate (3.1–4.2) | 140 (63.6) |
| | | High (≥ 4.3) | 13 (5.9) |
| | T3 | Low (≤ 3.0) | NR |
| | | Moderate (3.1–4.2) | 54 (60) |
| | | High (≥ 4.3) | NR |
| | T4 | Low (≤ 3.0) | NR |
| | | Moderate- High (> 3.0) | NR |
| PTS | T1 | None/Mild (≤ 25) | 349 (47.2) |
| | | Moderate/High (≥ 26) | 391 (52.8) |
| | T3 | None/Mild (≤ 25) | 48 (53.3) |
| | | Moderate/High (≥ 26) | 42 (46.7) |
| | T4 | None/Mild (≤ 25) | NR |
| | | Moderate/High (≥ 26) | NR |

Footnote: NR=not reported if n<10 in 1 or more categories. Since less than 10 participants experienced high burnout at T2 and T3, the moderate and high burnout categories were combined for reporting results.
*There were no participants scoring high on burnout at T4.

**Table 4. Changes in Burnout, Resilience, and PTS Across Time Using Repeated Measures ANOVA.**

| Measure | F | df | Partial Eta Squared | Significance (p) |
|---|---|---|---|---|
| Burnout Total | 2.63 | 2, 62 | .078 | .080 |
| Resilience | 2.53 | 2, 62 | .075 | .088 |
| PTS Total | 13.16 | 2, 62 | .298 | <.001 |

**Resilience.** At T2, nurses had a mean score of 3.4 on the BRS. The means at T3 and T4 were stable at 3.3. At T2, 30.5% of nurses met the cutoff for low resilience, which changed to 32.2% at T3 and 25% of the sample by T4. Resilience was not significantly affected by time, with $F (2, 62) = 2.53$, $p = .088$, though time explained 7.5% of the variance in changes in resilience, representing a medium effect size.

**PTS.** At T1 (May 2020 to June 2022), the mean score of the IES-R was 28.8. At T3, the mean score was 25.9, which decreased to 16.8 by T4. In addition, 52.8% of the sample met the cutoff for PTS at T1, which decreased to 46.7% by T3 (May 2021–2022), and further decreased to less than 25% in mid-2022 (T4). There were significant differences among the means of the post-traumatic stress scores over time, with $F (2, 62) = 13.16$, $p < .001$, and a large effect. Post hoc testing showed that differences in PTS were significant between T1 and T4 and T3 and T4 (Table 5).

## Path analysis using burnout and resilience as mediators between pandemic stress and PTS (Aim 2)

The results of the path analysis can be found in Fig 3. In the first regression analysis, pandemic stress was the outcome variable and work environmental factors and previous exposure to a traumatic event were modeled as the predictor variables, which explained about 23% of the variance in the model and were significant with $F (5, 84) = 6.4$ ($p < .001$) (Table 6).

Model 1 coefficients (Table 6) showed that only adequate PPE was significant in predicting pandemic stress ($p = .026$). The standardized beta weight of adequate PPE in Model 1 was small-to-moderate. For Model 2 with burnout as the outcome variable and pandemic stress as the predictor variable, pandemic stress explained approximately 15% of the variance in burnout and was significant with $F (1, 75) = 14.3$ ($p < .001$). The standardized beta was .40 ($p < .001$), showing a moderate-to-large association between burnout and pandemic stress. In Model 3, the outcome was PTS at T3 and the predictor was pandemic stress. The model was significant with $F (1, 75) = 11.92$ ($p < .001$). The tests of direct effects can be found in Table 7. When testing the indirect effects of burnout and resilience as mediators between pandemic stress and PTS, burnout was significant, but resilience was not; therefore, resilience was not found to serve as a mediator between pandemic stress and post-traumatic stress (Table 8). However, burnout acted as a mediator between pandemic stress and post-traumatic stress since the association between pandemic stress and post-traumatic stress was considerably reduced in size and no longer significant in the regression model with burnout included (see Fig 3).

**Table 5. RM-ANOVA Post hoc Testing for PTS Across Time.**

| Comparisons | Mean Difference | Standard Error | Significance* (p) |
|---|---|---|---|
| T1 vs. T3 | 4.88 | 2.61 | .214 |
| T1 vs. T4 | 11.63 | 2.02 | < .001 |
| T3 vs. T4 | 6.75 | 2.16 | .011 |

*Bonferroni adjustment for multiple comparisons.

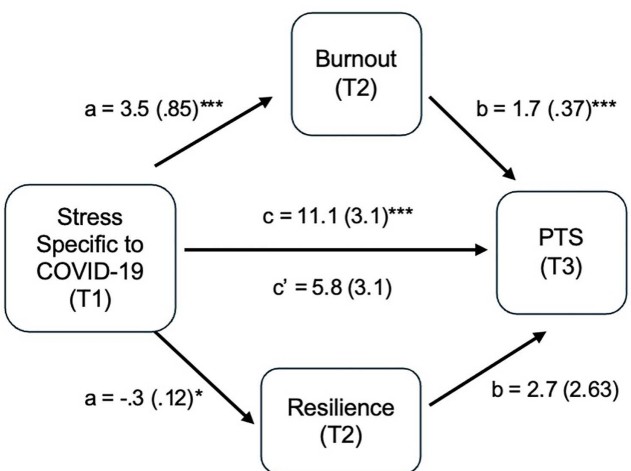

**Fig 3. Analysis of the Path Effects of Burnout and Resilience on Stress Specific to COVID-19 and PTS via Multiple Linear Regressions, with Standardized Beta Coefficients (Standard Error) Indicated in Model.** * $p<.05$, ** $p<.01$, *** $p<.001$.

**Table 6. Multiple Linear Regression of the Effects of Work Resources and Prior Traumatic Events on Stress Specific to COVID-19.**

| Predictor | Stress Specific to COVID-19 | | |
|---|---|---|---|
| | Standardized Beta | t | p- value |
| "Have you experienced other traumatic events prior to this experience with COVID-19?" | −.02 | −.15 | .879 |
| "Staffing levels are adequate for the current number of patients being treated in my workplace." | −.09 | −.60 | .548 |
| "We have adequate personal protective equipment to reduce risk of infection in my workplace." | −.29 | − 2.27 | .026 |
| "We have adequate ventilating devices and other lifesaving equipment to handle the current number of patients in my workplace." | −.08 | −.60 | .552 |
| "My workplace has made adequate preparations for a potential influx of COVID-19 patients in the future." | −.16 | −.98 | .333 |

**Table 7. Tests of Direct Effects of Burnout and Resilience on Stress Specific to COVID-19 and PTS at T3.**

| Path | B | SE | p − value | 95% CI | |
|---|---|---|---|---|---|
| | | | | Lower Level CI | Upper Level CI |
| Stress Specific to COVID-19 -- Burnout (path a) | 3.50 | .85 | < .001 | 1.80 | 5.19 |
| Burnout -- PTS (path b) | 1.69 | .37 | < .001 | .95 | 2.43 |
| Stress Specific to COVID-19 -- Resilience (path a) | −.25 | .12 | .041 | −.49 | −.01 |
| Resilience -- PTS (path b) | 2.72 | 2.63 | .304 | − 2.51 | 7.95 |
| Stress Specific to COVID-19 -- PTS (path c) | 11.07 | 3.09 | < .001 | 4.94 | 17.21 |
| Pandemic Stress -- Burnout & Resilience -- PTS (path c') | 5.85 | 3.06 | .060 | −.24 | 11.94 |

**Table 8. Tests of the Indirect Effects Using Bootstrapping.**

| Path | B | SE | 95% CI | |
|---|---|---|---|---|
| | | | Lower Level CI | Upper Level CI |
| Pandemic Stress -- Burnout -- PTS | 5.91 | 1.44 | 3.14 | 8.91 |
| Pandemic Stress -- Resilience -- PTS | −.68 | .84 | − 2.61 | .90 |

## Discussion

The purpose of this study was to explore how resilience, burnout, and post-traumatic stress (PTS) changed among hospital-based nurses during the COVID-19 pandemic. In this secondary analysis of a longitudinal study, PTS symptoms were highest during the first year (T1; 52.8%), with a significant decrease by the two-year follow-up (T4; < 25%). Notably, PTS at T1 strongly predicted PTS at the third time point (T3), emphasizing the need for early mental health interventions among healthcare workers. Burnout was also identified as a mediator between pandemic stress and PTS, suggesting that the impact of pandemic stress on PTS was intensified in the presence of high burnout.

Our findings contribute to the growing body of literature on nurses' mental health during the pandemic. Similar trends were observed in the Healthcare Worker Exposure Response & Outcomes (HERO) study, where 27.1% of nurses experienced PTSD in mid-2022, which was slightly higher than what was found in our sample (22%) during a similar time period [43]. Likewise, Cai et al. reported that 33% of nurses in China experienced PTS during the January 2020 outbreak, which decreased to 19% by February 2020 [11]. These findings, along with ours, highlight how PTS symptoms were most pronounced early in the pandemic and declined over time.

In our study, nurse burnout remained relatively stable from 2020 to 2022, with up to 47% reporting moderate burnout and 38% reporting low resilience. This aligns with mixed evidence in the existing literature: some studies found stable burnout levels, while others reported decreases. For example Aiken et al. documented pre-pandemic nurse burnout rates as high as 54%, with no significant change during the pandemic [44]. Other cross-sectional studies reported burnout prevalence rates ranging from 34.1% to 69% during the pandemic, [45,46] and large national datasets from the United States such as NSSRN, ANF *Pulse on the Nation Series*, and Aiken et al. showed rates between 51–60% [44,47,48]. Notably, many studies showing changes in burnout over time did not follow the same participants longitudinally [12,15]. In contrast, our longitudinal data support other's findings that high burnout levels predated the pandemic and reinforce the need for healthcare leaders to address these longstanding issues to prevent further workforce attrition [44,49].

Using the adapted CTS model, we also found that inadequate access to personal protective equipment (PPE) contributed to increased pandemic-related stress. In turn, baseline pandemic stress was associated with higher PTS symptoms at one-year follow-up. These findings align with previous research showing that inadequate work resources, such as insufficient PPE, are associated with greater psychological distress and intent to leave the profession [42]. Similarly, the Health Resources and Services Administration has identified unsafe working conditions and insufficient resources as key contributors to nurse burnout and turnover [48]. Participants in our study reported better access to PPE and lower staffing levels later in the pandemic, possibly reflecting ongoing nursing turnover, supply chain disruptions, and national stockpile deficiencies [48,50]. The inadequacy of the U.S. PPE stockpile and hospital procurement models exacerbated early shortages, intensifying stress among frontline workers [50]. To prepare for future pandemics, healthcare systems must ensure sufficient emergency equipment, PPE, and staffing capacity. Contingency planning should prioritize scalable workforce models and resilient supply chains to meet increased patient volume and acuity.

While burnout rates in our sample were slightly lower than those reported in large, national datasets, such as NSSRN, ANF, and others, [44,47,48] our prevalence of PTS was lower than what was reported in the HERO study (< 22%, n = 32 vs. 27.1%, n = 550, respectively) [43]. Although we did not assess nurse turnover directly, our findings reinforce those of national reports, which consistently cite burnout, inadequate staffing, and unsafe conditions as top reasons for attrition [47,48]. Healthcare leaders should act urgently to support the mental well-being of nurses, particularly during the early and most intense phases of a crisis. Proactive interventions to reduce pandemic stress, such as ensuring sufficient staffing and access to protective resources, may help mitigate long-term psychological impacts.

Additionally, the pandemic revealed critical gaps in the national PPE supply chain and budgeting models [50]. Current PPE procurement strategies often prioritize cost savings over safety, leaving frontline workers vulnerable [50]. Moving forward, healthcare systems must restructure PPE budgeting to ensure reliable and adequate supplies, and federal agencies should invest in rebuilding and maintaining a national stockpile that includes safe, unexpired, and well-distributed

emergency equipment [50]. These investments are not only vital for protecting healthcare workers but may also help retain the workforce during and after future health crises. Finally, mental health support must be prioritized at the onset of future pandemics, with sustained services for those who experienced prolonged distress or trauma.

### Limitations

There are several limitations of this study due to the nature of it being a secondary data analysis. Demographic data were only collected at the time of the first questionnaire. Therefore, we do not know if participants changed jobs during the study period. These changes may have influenced their resilience, burnout, and post-traumatic stress. In addition, the initial enrollment period for the baseline survey was 13 months, and the survey at T2 had some overlap with T1 from November 2020 to May 2021, depending on when the participant enrolled in the study. The start of the COVID-19 pandemic involved rapidly changing protocols and shortages of hospital beds, equipment, and PPE. To address this, we compared the findings of participants who responded to T1 during the first half of the enrollment period and those who joined the study in the second half of T1. There were no significant differences in mental health outcomes between the groups.

Overall, approximately 95% of the sample dropped out between T1 and T4, with the highest attrition rate (70%) occurring between T1 and T2. Some potential reasons for attrition might have been a lack of financial incentive to participate in the study and survey fatigue due to increased surveying of nurses over the course of the pandemic by hospitals, researchers, and professional organizations. Anecdotally, the researchers in the primary study who analyzed qualitative CHAMPS data found that many participants wanted to move on from the pandemic and put it behind them, which hindered their continued participation in the longitudinal study. Generalizability was limited due to the small sample size and that more than half of the sample resided in the northeast. Nonetheless, this study fills a gap in the science regarding the longitudinal effects that working during the COVID-19 pandemic had on U.S. nurses' well-being.

### Conclusions

This study was the first to examine the longitudinal effects of burnout, resilience, and PTS in U.S. nurses during the COVID-19 pandemic from 2020 to 2022. One of the major strengths of this study is that it analyzed multiple data collection points over a two-year period, which is unique since many studies conducted during the COVID-19 pandemic were cross-sectional in nature or only collected longitudinal data for one-year or less. This study was the first to evaluate mediation of pandemic stress to post-traumatic stress longitudinally during the COVID-19 pandemic. Further research is needed to determine if PTS and burnout have persisted as the rates of COVID-19 infections continue to fluctuate across the United States. In addition, studies of interventions focused on improving work environmental resources, such as PPE, should be conducted to evaluate their longitudinal effects on healthcare worker mental health and retention.

### Supporting information

**S1 Table. Mauchly's Sphericity Assumption Results for Aim 1.**
(DOCX)

**S2 Table. Comparison of Participant Responses that Enrolled Early vs. Late at T1 to Work Adequacy Items, Stress Specific to COVID-19, PTS, Burnout, and Resilience.**
(DOCX)

**S3 Table. Description of Participant Characteristics Who Remained in the CHAMPS Study Over Time Using Chi Square Analysis. NR = Not Reported due to n < 10.**
(DOCX)

## Acknowledgments

There are several people who played an integral role in the completion this study. Dr. Mary Ann Cantrell's support and ideas helped bring this secondary analysis to fruition. Additional appreciation is extended to Dr. Mary Ann Heverly and Dr. Margaret Brace, the statisticians for this study, and the CHAMPS Steering Committee, without whom the primary study would not exist. Thank you to everyone who participated in the CHAMPS Study and to all the frontline healthcare workers who cared for patients throughout the COVID-19 pandemic.

## Author contributions

**Conceptualization:** Rachel G. Baskin, Linda C. Copel, Janell L. Mensinger, Heather Brom.

**Data curation:** Janell L. Mensinger.

**Formal analysis:** Rachel G. Baskin.

**Funding acquisition:** Rachel G. Baskin.

**Investigation:** Linda C. Copel, Janell L. Mensinger, Heather Brom.

**Methodology:** Rachel G. Baskin, Janell L. Mensinger.

**Writing – original draft:** Rachel G. Baskin.

**Writing – review & editing:** Linda C. Copel, Janell L. Mensinger, Heather Brom.

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
