## [Decision Letter · Decision Letter 0]

21 Feb 2025

PONE-D-24-59485Nurse Resilience, Burnout, Work Stress, and Post-Traumatic Stress During COVID-19: A Secondary Longitudinal AnalysisPLOS ONE

Dear Dr. Baskin, 

Thank you for submitting your manuscript to PLOS ONE. After careful consideration, we feel that it has merit but does not fully meet PLOS ONE’s publication criteria as it currently stands. Therefore, we invite you to submit a revised version of the manuscript that addresses the points raised during the review process. Please submit your revised manuscript by  Apr 07 2025 11:59PM. If you will need more time than this to complete your revisions, please reply to this message or contact the journal office at plosone@plos.org. Please include the following items when submitting your revised manuscript:

We look forward to receiving your revised manuscript.

Kind regards,

Majed Sulaiman Alamri, PhD

Academic Editor

PLOS ONE

Journal Requirements:

“The costs associated with this secondary study were partially funded by Villanova University’s Alpha Nu Chapter of Sigma Theta Tau International. The primary study was supported by grants from Travere Therapeutics Inc, McKesson Corporation, anonymous donors, and internal funding from the M. Louise Fitzpatrick College of Nursing at Villanova University.”

4. In the online submission form, you indicated that [Data Availability: Deidentified CHAMPS data and codebooks are available upon request from the Villanova University Fitzpatrick College of Nursing. Please contact the study PI, Dr. Margaret Brace (margaret.brace@villanova.edu).].

Reviewers' comments:

Reviewer's Responses to Questions

**Comments to the Author**

1. Is the manuscript technically sound, and do the data support the conclusions?

Reviewer #1: No

Reviewer #2: Yes

2. Has the statistical analysis been performed appropriately and rigorously? 

Reviewer #1: Yes

Reviewer #2: Yes

3. Have the authors made all data underlying the findings in their manuscript fully available?

Reviewer #1: Yes

Reviewer #2: Yes

4. Is the manuscript presented in an intelligible fashion and written in standard English?

Reviewer #1: Yes

Reviewer #2: Yes

5. Review Comments to the Author

Reviewer #1: This paper is a good one in its field & I have made some comments on the manuscript that I hope can strengthen the quality of the paper:

1- The Abstract and whole article should be organized and should be rewritten based on scientific writing.

2- There are no references for some claims and sentences in the text.

3-Please mention the sampling method

4- Please mention how the validity & reliability of the research were examined.

5- Discussion is very poor. Please revise it. In discussion you should discuss about your results with base of knowledge and related literature & references.

6- The limitation part should be more complete and the authors should mention how they overcome.

7- Overall, It is found that the paper is difficult to follow due to consistency & coherency in writing. I suggest the authors scale back the aims of research, questions, analysis, findings, discussion of this paper to provide more space to introduce the concepts, findings etc.

11- The manuscript should be revised by a native English people.

Reviewer #2: The study provides valuable insights into nurse well-being during the pandemic, but the small sample size, lack of significant changes in burnout and resilience, and missing analysis of individual differences leave key questions unanswered. A larger study with a more diverse sample and additional contextual factors could provide a clearer picture. Despite reporting a decline in post-traumatic stress (PTS), burnout and resilience did not significantly change over time.Despite reporting a decline in post-traumatic stress (PTS), burnout and resilience did not significantly change over time. Given the known stressors of the pandemic, it is surprising that burnout did not change, and this contradiction is not well-explained in the discussion.

6. PLOS authors have the option to publish the peer review history of their article (what does this mean?). If published, this will include your full peer review and any attached files.

Reviewer #1: No

Reviewer #2: No

---

## [Author Response · Author response to Decision Letter 1]

30 May 2025

Dear Dr. Alamri and Reviewers:

Thank you for this opportunity to revise and resubmit our submission [PONE-D-24-59485] to PLOS One. We greatly appreciate the feedback from the reviewers as well as your suggestions. We have addressed each of the reviewer’s comments in the table below and provided the location where the corrections may be found in the revised manuscript.

Academic Editor

Thank you for this feedback and for providing resources. We have adjusted our manuscript according to the journal’s style requirements.

We note that the grant information you provided in the ‘Funding Information’ and ‘Financial Disclosure’ sections do not match.

The funding for both the primary study and secondary analysis were from private funders, which did not include grant numbers.

Thank you for stating in your Funding Statement:

“The costs associated with this secondary study were partially funded by Villanova University’s Alpha Nu Chapter of Sigma Theta Tau International. The primary study was supported by grants from Travere Therapeutics Inc, McKesson Corporation, anonymous donors, and internal funding from the M. Louise Fitzpatrick College of Nursing at Villanova University.”

We have provided an amended statement in the cover letter and have added it here for convenience:

This work received funding from Villanova University’s Falvey Memorial Library Scholarship Open Access Reserve (SOAR) Fund and Villanova University’s Alpha Nu Chapter of Sigma Theta Tau International. There was no additional external funding received for this study. The primary study was supported by grants from Travere Therapeutics Inc, McKesson Corporation, anonymous donors, and internal funding from the M. Louise Fitzpatrick College of Nursing at Villanova University.

In the online submission form, you indicated that [Data Availability: Deidentified CHAMPS data and codebooks are available upon request from the Villanova University Fitzpatrick College of Nursing. Please contact the study PI, Dr. Margaret Brace (margaret.brace@villanova.edu).].

Thank you for this information. The data cannot be made publicly available due to risk of confidentiality and a breach of compliance with our institution’s IRB. Data are currently available by application only. Therefore, we would like to submit an exemption request. We have amended our data availability statement as follows:

Data cannot be shared publicly because of the risk of loss of confidentiality to participants. Data are available from the Villanova University Institutional Review Board for researchers who meet the criteria for access to confidential data. Deidentified CHAMPS data and codebooks are available upon request from the Villanova University Fitzpatrick College of Nursing. Please contact the study PI, Dr. Margaret Brace (margaret.brace@villanova.edu).

Please include captions for your Supporting Information files at the end of your manuscript, and update any in-text citations to match accordingly. Please see our Supporting Information guidelines for more information: http://journals.plos.org/plosone/s/supporting-information.

Thank you for this resource. We have updated the Supporting Information files according to the guidelines.

Reviewer #1:

Reviewer #1: This paper is a good one in its field & I have made some comments on the manuscript that I hope can strengthen the quality of the paper:

1- The Abstract and whole article should be organized and should be rewritten based on scientific writing.

The abstract and article have been formatted to meet the requirements of PLOS One (https://journals.plos.org/plosone/s/submission-guidelines).

2- There are no references for some claims and sentences in the text.

We appreciate the feedback from this reviewer. We have reviewed the manuscript to ensure that statements were appropriately supported with citations.

METHODS 3-Please mention the sampling method.

The sampling method was described in the methods section (page 9, line 308), stating “The CHAMPS study was conducted in the U.S. using snowball sampling and advertising via local and national professional organizations.” We added an additional statement about providing the study link on social media platforms.

METHODS Continued 4- Please mention how the validity & reliability of the research were examined.

Our manuscript contains information about the reliability of all instruments from previous studies, as well as reporting the Cronbach alpha value from this study. External validity was addressed in the limitations section by discussing generalizability. Internal validity, such as the attrition rate, were addressed in the limitations section. If there is a specific aspect of validity and reliability that remains of concern, please let us know.

DISCUSSION 5- Discussion is very poor. Please revise it. In discussion you should discuss about your results with base of knowledge and related literature & references.

We revised the discussion section using the PLOS suggested guidelines found here: https://plos.org/resource/how-to-write-conclusions/. We focused the initial paragraph on the findings of our study. The second paragraph focuses on what this study adds to the literature and compares our findings to other studies. Due to our small sample size, we found it important to compare our findings to those of larger datasets.

LIMITATIONS 6- The limitation part should be more complete and the authors should mention how they overcome.

In the limitations section, we have added a statement about how we overcame our study’s limitations, such as the 13-month enrollment period of T1. This statement can be found on pages 26, lines 482-488. When we compared the mental health outcomes of those who responded in the first half of T1 to the second half, there were no significant differences in burnout, resilience, or PTS. Due to the nature of this being a secondary study, we did not have the ability to control the sample size or timing of measurements of interest.

GENERAL 7- Overall, It is found that the paper is difficult to follow due to consistency & coherency in writing. I suggest the authors scale back the aims of research, questions, analysis, findings, discussion of this paper to provide more space to introduce the concepts, findings etc.

Given the length of the paper, we scaled back the introduction section, refined the description of our findings and enhanced the discussion. We also removed research aim 2 to better highlight how our study contributes new findings to the literature. Although there was only enough power in the mediation analysis for a medium-to-large effect, we chose to report the findings of that research aim because some relationships within the model were still found to be significant. We also addressed the sample size and attrition in the limitations section on page 27, lines 489-498.

11- The manuscript should be revised by a native English people.

Thank you for your feedback. We understand your concerns regarding language and phrasing. We have carefully reviewed the manuscript for clarity and grammatical accuracy and worked with an editor to improve the consistency and coherency.

REVIEWER 2 Reviewer #2: The study provides valuable insights into nurse well-being during the pandemic, but the small sample size, lack of significant changes in burnout and resilience, and missing analysis of individual differences leave key questions unanswered. A larger study with a more diverse sample and additional contextual factors could provide a clearer picture. Despite reporting a decline in post-traumatic stress (PTS), burnout and resilience did not significantly change over time. Given the known stressors of the pandemic, it is surprising that burnout did not change, and this contradiction is not well-explained in the discussion.

Since this study used pre-existing data, we are unable to change the sample size. However, the sample size was large enough to power this research aim. An additional factor that should be considered, which has been added to the discussion section of the manuscript on page 24, lines 427-430, is that burnout rates among nurses in the United States were already high prior to the pandemic. According to work by Aiken et al. (2023) that included 40,674 nurses, burnout rates did not change significantly pre- to post-COVID-19 pandemic. Although Aiken et al. (2023) used a repeated cross-sectional study, their sample size was large and speaks to the increased rates of burnout that likely existed before the pandemic started.

Analyzing individual nurse differences would be an interesting addition but is beyond the aims of the study. We assessed for differences in outcomes in those who enrolled in the first 6-months of the study with those who enrolled in the second half of the enrollment period (T1) due to the rapid nature of the pandemic as it evolved. Analyzing differences in resilience, PTSD, and burnout based on participant demographics was already studied in the COVID-19 health workforce literature (e.g., Alfonsi et al., 2023; Armstrong et al., 2022; Slykerman et al., 2021),and therefore, we did not include this in our study aims.

---

## [Decision Letter · Decision Letter 1]

10 Jul 2025

Nurse resilience, burnout, pandemic stress, and post-traumatic stress: A secondary analysis of a longitudinal cohort

PONE-D-24-59485R1

Dear Dr. Baskin, 

We’re pleased to inform you that your manuscript has been judged scientifically suitable for publication and will be formally accepted for publication once it meets all outstanding technical requirements.

Kind regards,

Majed Sulaiman Alamri, PhD

Academic Editor

PLOS ONE

Additional Editor Comments (optional):

Reviewers' comments:

Reviewer's Responses to Questions

**Comments to the Author**

1. If the authors have adequately addressed your comments raised in a previous round of review and you feel that this manuscript is now acceptable for publication, you may indicate that here to bypass the “Comments to the Author” section, enter your conflict of interest statement in the “Confidential to Editor” section, and submit your "Accept" recommendation.

Reviewer #1: All comments have been addressed

2. Is the manuscript technically sound, and do the data support the conclusions?

Reviewer #1: Yes

3. Has the statistical analysis been performed appropriately and rigorously? 

Reviewer #1: Yes

4. Have the authors made all data underlying the findings in their manuscript fully available?

Reviewer #1: Yes

5. Is the manuscript presented in an intelligible fashion and written in standard English?

Reviewer #1: Yes

6. Review Comments to the Author

Reviewer #1: (No Response)

7. PLOS authors have the option to publish the peer review history of their article (what does this mean?). If published, this will include your full peer review and any attached files.

Reviewer #1: No

---

## [Editor Report · Acceptance letter]

PONE-D-24-59485R1

PLOS ONE

Dear Dr. Baskin,

I'm pleased to inform you that your manuscript has been deemed suitable for publication in PLOS ONE. Congratulations! Your manuscript is now being handed over to our production team.

Kind regards,

on behalf of

Prof. Majed Sulaiman Alamri

Academic Editor

PLOS ONE